

# Geophysical evidence of pre-sag rifting and post-rifting fault reactivation in the Parnaíba basin, Brazil

D. L. de Castro[1], F. H. R. Bezerra[1], R. A. Fuck[2], Roberta M. Vidotti[2]

[1]Programa de Pós-Graduação em Geodinâmica e Geofísica, Universidade Federal do Rio Grande do Norte, Natal, 59078-970, Brazil
[2] Instituto de Geociências, Universidade de Brasília, Campus Universitário, Brasília, 70910-900, Brazil

*Correspondence to*: D. L. de Castro (david@geologia.ufrn.br)

**Abstract.** This study investigated the rifting mechanism that preceding the prolonged subsidence of the Paleozoic Parnaíba basin in Brazil and shed light in the tectonic evolution of this large cratonic basin in the South American platform. From the analysis of aeromagnetic, aerogravity, reflection seismic and borehole data, we concluded the following: (1) Large pseudo-gravity and gravity lows mimic graben structures but are associated with linear supracrustal strips in the basement. (2) Seismic data indicate that rift zones 120-200 km wide and up to 300 km long occur in other parts of the basins. These rift zones mark the early stage of the 3.5-km-thick sag basin. (3) The rifting phase occurred in the Early Paleozoic and had a subsidence rate of 47 m/Myr. (4) This rifting phase was followed by a long period of sag basin subsidence at a rate of 9.5 m/Myr between the Silurian and the Late Cretaceous, during which rift faults propagated and influenced deposition. These data interpretations support the following succession of events: (1) After the Brasiliano orogeny (740-580 Ma), brittle reactivation of ductile basement shear zones led to normal and dextral oblique-slip faulting concentrated along the Transbrasiliano Lineament, a continental-scale shear zone that marks the boundary between basement crustal blocks. (2) The post-orogenic tectonic brittle reactivation of the ductile basement shear zones led to normal faulting associated with dextral oblique-slip crustal extension. In the west, the orthogonal fault-inducing rifting resulted in pure-shear extension, producing rift zones that crosscut metamorphic foliations and shear zones within the Parnaíba block. (3) The rift faults experienced multiple reactivation phases. (4) Similar processes may have occurred in coeval basins in the Laurentia and Central African blocks of Gondwana.

## 1 Introduction

The transition between the Late Neoproterozoic and Early Cambrian was marked by the final assembly of West Gondwana via closure of the Brasiliano/Pan-African ocean basins, amalgamation of cratonic fragments and incorporation of accretionary complexes into mobile belts (Dalziel, 1997; Oliveira and Mohriak, 2003; Cordani et al., 2013). The closure of the Goiás-Pharusian ocean sutured the Laurentian-Amazonian-West African and Gondwanan Central African blocks along



the Transbrasiliano-Kandi megashear zone and, secondarily, the Araguaia suture zone (Cordani et al., 2013; Stampfli et al., 2013; Brito Neves and Fuck, 2014) (Fig. 1).

Widespread post-orogenic extension occurred, separating Laurentia and Baltica from the proto-Andean margin of West Gondwana and opening the Iapetus Ocean (Bond et al., 1984; Dalziel, 1997). During the Late Cambrian–Early Ordovician,

changes in the stress state of the continental lithosphere caused crustal extension and rifting, which were occasionally accompanied by abundant intrusions (Stampfli et al., 2013). The rifting and magmatic episodes occurred primarily along regional lithospheric anisotropies in West Gondwana and Laurentia, and these episodes did not result in continental breakup. Early Paleozoic cratonic basins formed on the failed rifts and occupied large regions of North America (e.g., the Michigan, Illinois, and Hudson Bay basins), Africa (e.g., the Congo, Chad, and Taoudeni basins), and South America (e.g., the

Amazonas, Parecis, Paraná, and Parnaíba basins) (Hartley and Allen, 1994; Armitage and Allen, 2010).

The Parnaíba basin is one of the largest cratonic basins in the West Gondwanan South American platform. This roughly circular basin is broadly saucer-shaped and featured relatively slow, long-lived subsidence events (~355 Myr in duration) from the Silurian to the Late Cretaceous (Góes and Feijó, 1994). Preceding the sag sedimentation, fault reactivation of the Brasiliano shear zones controlled the formation of a set of rift basins. Therefore, changes in the stress state and thermal

structure of the lithosphere along the Transbrasiliano megashear zone induced motion of anomalous masses in the lithosphere and/or asthenosphere during this extensional event and driving the long-term thermal subsidence of the Parnaíba basin. According to De Rito et al. (1983) and Sloss (1990), movement of anomalous mass is the most successful mechanism associated with the formation of cratonic basins.

However, several uncertainties regarding the generation of the Parnaíba basin remain. First, the mechanism that

generated accommodation space in the large Paleozoic Parnaíba basin is still not completely understood. Over the years, researchers have proposed that the onset of the long-term thermal subsidence of the cratonic basin was preceded by a rifting process, which terminated in the Ordovician (Brito Neves et al., 1984; Cordani et al., 1984). However, the driving mechanisms, which are responsible for the accommodation space in the basin, rifting and thermal subsidence, and the relative timing of these events are not clear. In addition, after the Brasiliano orogeny, a tectonic inversion occurred in West

Gondwana, generating linear grabens controlled by the Precambrian structures (metamorphic foliations and ductile shear zones), such as the Jaibaras basin (Fig. 2). De Castro et al. (2014) proposed a rifting stage preceding the Jaibaras graben based on gravity and magnetic anomalies and on borehole data reported by Cordani et al. (1984). However, the existence of this ancient rifting system remains speculative due to a lack of direct evidence from seismic data and exploratory wells.

The second uncertainty is associated with the geophysical signature of the basin and the interpretation of its architecture.

The Early Paleozoic extensional episode formed a complex system of aborted rift basins, which are partially exposed at the boundaries of the Parnaíba basin. The Jaibaras trough is the primary outcropping example of these rift basins (Oliveira and Mohriak, 2003) (Fig. 2). The location of these basins appears to be restricted to the Brasiliano shear zones. Nevertheless, previous geophysical data, seismic sections and well logs suggest that the majority of the Parnaíba basin is underlain by graben-like structures (Nunes, 1993; Cordani et al., 2009). More recently, De Castro et al. (2014) proposed a more accurate



distribution of the rifts beneath the Parnaíba basin. Based primarily on gravity and magnetic anomaly patterns, they describe two distinct rifting stages between the Late Neoproterozoic and the Early Paleozoic, preceding deposition in the major cratonic sag. These authors mapped the concealed grabens using general gravity minima and low-amplitude magnetic anomalies to identify the depocenters of sedimentary basins. However, this premise can be easily violated where causative

sources within a heterogeneous basement interfere with the potential field anomalies. Thus, independent geophysical and geological data are often used to constrain the gravity and magnetic models.

      The third uncertainty involves the broad implications of the evolution of the Parnaíba basin, which still needs to be assessed and compared to other basins in South America and Africa. The view of sag basin evolution in the Paleozoic has led many studies to overlook the possibility of multiple rift mechanisms in other sedimentary basins.

A previous study by De Castro et al. (2014) attempted to map the complex lithospheric structure beneath the Parnaíba basin, using solely airborne and satellite magnetic and gravity data. They proposed two rift stages before sag sedimentation, but that model needed confirmation by more specific geophysical and/or geological information. Airborne geophysical data are used worldwide to map large-scale crustal structures (e.g., Nabighian et al., 2005; Grauch and Hudson, 2007; Anudu et al., 2014), especially in large sedimentary basins, such as the Parnaíba Basin (670,000 km2). To minimize the reduced

resolution of airborne data for mapping the basin internal geometry, we introduced seismic and well data, which are the most appropriate geophysical methods for investigations of basin architecture and served to reduce ambiguity in the interpretation of magnetic and gravity data. In the present study, however, we interpret the potential field maps on the basis of new seismic sections and well logs to determine the rift distribution, structural inheritance, and internal geometry of the Parnaíba basin (Fig. 2). Furthermore, 2D gravity and magnetic joint modeling along the seismic sections allowed us to understand how the

basement heterogeneities interfere with the potential field anomalies. Based on the new seismic data and geophysical models, we recognized at least three tectonic styles underlying the sag basin. These styles are related to the type and kinematics of faults (normal, strike-slip, reverse or thrust) and intensity of deformation (number and amount of fault offset and size of faults and folds). These tectonic styles have an influence on the final basin architecture. We discuss the influence of the Eopaleozoic rifting process in the prolonged periods of subsidence experienced by the Parnaíba cratonic basin and the

mechanisms for its formation. Finally, we assess the broad implications for other sedimentary basins in South America and West Africa.

## 2 Geological Setting

### 2.1 General features and main sedimentary-volcanic units

The Parnaíba basin is a large Paleozoic cratonic basin located in northeastern South America (Fig. 2). This basin is ~1,000

30     km long and ~970 km wide and occupies a complex region of West Gondwana, which was subjected to crustal collisions involving cratonic blocks, extensive fold belts and concealed basement inliers (Parnaíba block). The basement area stabilized



when the Late Proterozoic Brasiliano/Pan-African orogeny (720-540 Ma) ceased (Cordani et al. 1984, 2009; De Castro et al., 2014).

The evolution of the Parnaíba basin involved five primary tectono-sedimentary sequences, two magmatic pulses, and an area that was at least five times larger than that affected by the aborted rifts (Fig. 3 and Table 1). The present-day sag basin consists of at least four tectono-sedimentary sequences separated by regional unconformities, comprising distinct deposition history (Góes and Feijó, 1994). The sedimentary fill in the sag basin is up to 3.5 km thick and is primarily composed of Early Silurian to Cretaceous continental sediments in the form of several unconformity-bounded packages (Table 1). Recent seismic sections reported by the Brazilian Petroleum Agency (Alves, 2012; Ferreira, 2013) reveal grabenlike structures beneath the sag sequences that have been filled with ~3.0 km of pre-Silurian sequences.

The basin is composed of thick, primarily siliciclastic, epicontinental sequences (Góes and Feijó, 1994). Sequences I to V are the Cambrian-Ordovician (Jaibaras Group), Silurian-Early Devonian (Serra Grande Group), Middle Devonian-Early Carboniferous (Canindé Group), Late Carboniferous-Early Triassic (Balsas Group), and Jurassic-Cretaceous (Fig. 3 and Table 1).

## 2.2 Main tectonic features

Continental-scale shear zones (lineaments) played a major role in the Brasiliano orogeny and in the evolution of the Parnaíba Basin. These shear zones mark sutures associated with continental collisions such as the Araguaia and Transbrasiliano lineaments (Fig. 2). The 1,000-km-long Araguaia suture zone represents the final Neoproterozoic collision between the Amazonian craton, overlain by the allochthonous Araguaia belt, and the pre-Neoproterozoic Parnaíba block (Brito Neves and Fuck, 2014). Another important shear zone is the Transbrasiliano lineament. Many studies considered the Transbrasiliano lineament to be a continental-scale discontinuity characterized by strong long-wavelength magnetic anomalies and by low S wave velocities in the mantle (e.g., Fairhead and Maus 2003; Feng et al., 2004; Fuck et al., 2008; Brito Neves and Fuck, 2014). On the NE side of the Parnaíba basin margin, the Transbrasiliano lineament separates two Neoproterozoic crustal domains of the Borborema province (Médio Coreaú and Ceará Central; Fig. 2). The Transbrasiliano Lineament also controlled the internal rift geometry and formed a 150-km-wide rift zone in the eastern portion of the basin. Later reactivations of the Brasiliano shear zones deformed post-rift sequences, including post-Devonian tectonic inversion (Destro et al., 1994). These lineaments also form Precambrian lithospheric-scale boundaries. They were identified in a deep crustal, seismic reflection profile across the Parnaíba basin (Daly et al., 2014) and represent the collisional sutures of the Amazonian and the São Francisco cratons (De Castro et al., 2014).

Following the Brasiliano/Pan-African orogeny, tectonic inversion generated elongated grabens controlled by Precambrian structural fabric, which is mainly marked by ductile shear zones in the basement. The best examples of these grabens are the Jaibaras basin and other smaller Cambrian-Ordovician rift basins that are partially exposed at the northern and eastern edges of the Parnaíba basin (Fig. 2). The Jaibaras is the best known of these basins. It crops out at the NE



boundary of the Parnaíba basin, forms a NE-trending, 120-km-long and 10-km-wide graben generated by the reactivation of the Transbrasiliano lineament in the Cambrian-Ordovician (Fig. 2) basin (Oliveira and Mohriak, 2003).

## 3 Mapping Cambrian-Ordovician grabens

In sedimentary basins, gravity and magnetic lows are generally related to depocenters because the sedimentary fill is less

dense and less magnetic than crystalline basement rocks. Based on this assumption, De Castro et al. (2014) identified magnetic and gravity lows, which they interpreted as graben-like structures buried by the sag units of the Parnaíba basin. These authors recognized a direct correlation between a series of Cambrian-Ordovician grabens, partially exposed at the E and SW edges of the basin (Fig. 2), and negative pseudo-gravity anomalies. Thus, according to their interpretation, two sets of troughs occurred in the basin, particularly along the Brasiliano shear zones in the eastern portion of the basin. Two rift

systems were active prior to the widespread sag deposition, likely in the Late Neoproterozoic and Early Paleozoic (Nunes, 1993; Cordani et al., 2009; De Castro et al. 2014).

However, the premise that magnetic and gravity lows indicate buried rifts can be easily violated by causative sources within the basement (Blakely, 1996; De Castro et al., 2007). The residual magnetic and gravity anomalies, after extraction of the long-wavelength regional component associated with the deeper crustal structure, depend on the combined effects of the

basement units and the internal geometry of the sedimentary basin. In this study, we tested the hypothesis that negative anomalies represent either buried rifts or lithological units in the basement using two different approaches. The first approach consists of carefully analyzing the magnetic and gravity signatures of the outcropping Jaibaras basin and the Araguaia belt at the NE and W boundaries of the Parnaíba basin, respectively. The second approach consists of modeling potential field profiles constrained by new seismic data and well logs.

### 3.1 Airborne magnetic and gravity dataset

De Castro et al. (2014) compiled several airborne and satellite potential field datasets to determine the crustal domains beneath the Parnaíba basin. In this study, we focus on mapping the aborted rift system buried by the sag sedimentation using only the more recent aerogeophysical survey flown in 2005 and 2006 for the Brazilian Petroleum Agency (ANP). This airborne magnetic and gravity survey was conducted along E-W-oriented 6-km-spaced flight lines, with a 24-km tie-line

spacing in a N-S direction and a 1,100-m nominal flight height (Fig. 4). Data were recorded at intervals of 0.04 s (gravity) and 0.01 s (magnetic). ANP performed all necessary magnetic and gravity data reductions and leveling. The total magnetic field of the Earth was corrected for diurnal variations, the main geomagnetic field (IGRF), and leveling errors. The raw airborne gravity data correction involved vertical and horizontal accelerations, latitude, free-air reduction and Bouguer reduction. The resulting total magnetic intensity (TMI) and Bouguer anomaly data were interpolated onto 500-m and 1500-m

cell size grids, respectively, using bi-directional and minimum curvature methods, respectively. The bi-directional gridding technique was developed to cope specifically with the problem of different magnetic data density along and across flight



lines (Redford, 2006). Whilst, the minimum curvature interpolation algorithm fits a minimum curvature surface to the given data values nearest the coarse grid nodes (Briggs, 1974). This method is best for gravity data that have no dominant trend direction. The data processing routine, which was optimized by De Castro et al. (2014), was also applied to the current dataset to enhance short- to medium-wavelength negative magnetic and gravity anomalies directly associated with shallow

causative features within the upper crust. The processing stages include reduction to the magnetic pole (RTP), regional-residual separation and pseudo-gravity transformation for the residual magnetic data and regional-residual separation for the gravity data (Figs. 5a to c). The RTP transformation is a standard magnetic technique that corrects for shifts of anomalies from the centers of their magnetic sources related to the oblique orientation of the magnetic field with respect to Earth's surface (Blakely, 1996). The parameters of the RTP filter include a magnetic inclination of -11.16° and declination of -

21.26°, which we calculated for the central location of the aerogeophysical survey. A pseudo-inclination factor of 60° was used to provide a stable RTP computation in and near the declination direction at low magnetic latitudes (absolute inclination less than 20°). We removed the regional anomalies (wavelength more than 125 km) from the magnetic and gravity data (Figs. 5a, b) using a Gaussian filter with a standard deviation of the Gaussian function of 0.008 cycles/km as the spatial cutoff.

Baranov (1960) proposed a pseudo-gravity transformation to convert the total-field magnetic anomaly into the gravity anomaly that would be observed if the magnetization distribution were to be replaced with an identical density distribution (Blakely, 1996). Actually, the pseudo-gravity transformation converts the magnetic field to magnetic potential, leading the magnetic anomaly to a single edge response for each source border, making easier the interpretation of magnetic maps (Blakely and Simpson, 1986). This technique also reduces directional anisotropy of the magnetic response to the source

distribution, once the pseudo-gravity anomaly of a magnetic source is proportional to the magnetic potential of the same source with vertical magnetization (Blakely, 1996). In practice, we derived the pseudo-gravity transformation via an integration of the residual magnetic data reduced to the pole using a standard density of 2670 kg/m$^3$. Since this technique involves reduction to the pole, a same pseudo-inclination factor of 60° was applied to make the pseudo-gravity transformation stable. Finally, Figure 5d presents a new interpretative map, we highlight the low gravity and pseudo-gravity

anomalies associated with potential Cambrian-Ordovician grabens at the base of the Parnaíba basin.

  The magnetic and gravity maps presented in this study (Fig. 5) differ from those maps shown by De Castro et al. (2014) because these authors used several airborne magnetic surveys and gravity data derived from satellite, which are more comprehensive than the dataset described in this paper. In addition, De Castro et al. (2014) used an expanded dataset in order to investigate in a more regional scale the crustal domains that encompass the structural framework of the Parnaíba basin.

That dataset comprises several airborne magnetic surveys with different data acquisition setups (i.e., flight height, spacing between lines, flight direction), overflight areas and magnetometers. On the other hand, this study focused on the basin internal geometry. Thus, we opted to use the newest survey, whose measurements are distributed homogeneously throughout the entire basin and carried out with the same parameters and instruments.





In addition to the airborne surveys of the Parnaíba basin, we also included potential field data from the Jaibaras basin (Fig. 2). The Geological Survey of Brazil (CPRM) acquired high-resolution aeromagnetic data in 2009 with a flight line spacing of 500 m and a nominal height of 100 m. The data set was reduced to the magnetic pole (RTP), and the regional component of the geomagnetic field was subtracted to enhance the upper crustal structure (Fig. 6a). The gravity dataset

consists of 671 measurement stations from several terrestrial surveys conducted by Brazilian public universities and other research institutions. Although the gravity stations are sparse and irregularly distributed across the basin, the resulting Bouguer anomaly map provides an appropriate image of the regional-scale structures. The residual gravity values were obtained by applying the same Gaussian filter used for the airborne geophysical data (Fig. 6b).

## 3.2 Potential field data analyses

The difference between the magnetic and gravity patterns in the Parnaíba basin is the primary characteristic in the geophysical maps (Figs. 5a, b). The magnetic anomalies exhibit a NE-SW trend near the Transbrasiliano lineament and a more complex magnetic pattern to the west (Parnaíba block), with lineaments oriented E-W, N-S, and NW-SE. To the east (Borborema Province), extensive NE-oriented magnetic anomalies are related to the Brasiliano shear zones along the E and SW edges of the basin (E in Fig. 5a). In particular, the Transbrasiliano Lineament (TB) can be traced throughout the basin,

following a series of NE-trending magnetic anomalies (Figs. 5a, c). Interestingly, the Araguaia suture zone (F in Fig. 5) exhibits no marked trend in the magnetic data.

The residual gravity map presents a different anomaly pattern. The gravity lineaments trend N-S and curve to the east in the central part of the basin, where they are apparently anchored by the Transbrasiliano Lineament (G in Fig. 5b). This lineament appears to be an important crustal boundary because it separates areas with different gravity and magnetic

signatures, i.e., the Parnaíba block to the west and the Borborema province to the east. In the Parnaíba block, a series of N-S-elongated gravity lows and highs represent the Araguaia suture zone (F in Figs. 5b, d), as well as the northward extension of the Goiás magmatic arc beneath the western boundary of the Parnaíba basin (De Castro et al., 2014). Nevertheless, this feature is not evident in the magnetic anomaly map. In areas closer to the magnetic equator (absolute magnetic inclination less than 10°), the anomalies derived from N-S sources may not be detected. Because the geomagnetic field is horizontal in

this case, no magnetic boundaries are crossed along the north-south direction with respect to the east-west edges (Murthy 1998).

## 3.3 Magnetic and gravity anomaly patterns in the Jaibaras basin

We used the Jaibaras basin as an analogue of the graben system beneath the sag's sedimentary cover. Figure 6 presents the residual magnetic and gravity anomalies maps of the Jaibaras basin. Elongated positive magnetic and gravity anomalies

occur in the area of the Jaibaras basin. These anomalies are associated with surface and near-surface, moderate-to-high susceptibility, dense volcanic rocks within the basin and with basement units, particularly high-grade metamorphic rocks, such as granulites and eclogites, which crop out in both crustal domains (Cavalcante et al., 2003; Santos et al., 2009). The



high-amplitude anomalies related to basement causative sources conceal the relatively weak magnetic and gravity effects associated with the shallow sedimentary basin fill. In fact, the expected magnetic and gravity lows are observed outside the basin, where low-density granites and metasedimentary sequences are exposed. This geophysical setting can be projected to the Parnaíba basin. Rather than the graben systems proposed by previous studies (Cordani et al., 1984; Nunes, 1993), the

magnetic and gravity signatures in the Jaibaras basin suggest that lithological heterogeneities in the basement may be responsible for the negative anomalies in some degree.

Other clues along the western border of the Parnaíba basin can also be used to understand the sources of the gravity lows in the Parnaíba basin. A series of N-S-striking negative and positive anomalies occupy a 200-km-wide zone along the basin edge, parallel to the Araguaia suture zone (F in Fig. 5b). According to Ussami and Molina (1999), this region is related to the

northern extension of the Neoproterozoic Goiás magmatic arc, the easternmost branch of the Tocantins province (Fig. 2), beneath the Parnaíba basin. The westernmost gravity lows (H in Fig. 5b) correspond to low-grade metamorphic supracrustal sequences in the Araguaia belt, a marginal fold belt thrusted onto the Amazonian craton during the Brasiliano orogeny (Alvarenga et al., 2000). Thus, the gravity highs and lows reveal upper crustal undulations due to a collisional fold-and-thrust belt and lateral accretion associated with the closure of the Goiás-Pharusian Ocean between the Amazonian and São

Francisco cratons (Cordani et al., 2014). From this perspective, certain N-S-elongated and weakly arcuate gravity lows (G in Fig. 5b) could be associated with supracrustal terrains involved in the collision that formed West Gondwana during the Neoproterozoic-Early Paleozoic. The subsequent Cambrian-Ordovician rifting developed on this tectonic framework, creating a complex pattern of gravity anomalies.

## 3.4 Seismic and well dataset

In this study, we introduced seismic reflection data and well logs, which were acquired in the 1970s, 1980s and in recent years (2006-2010) by the Brazilian Oil Agency (ANP). ANP provided the seismic sections already processed and time migrated, which revealed the tectono-sedimentary sequence and basin architecture. We also used these seismic sections and well logs to constrain the interpretation and modeling of the gravity and magnetic data. We analyzed ten 2D stacked and migrated seismic profiles to map key reflections and discontinuities using standard seismic sequence stratigraphic criteria

and lithological calibration derived from two well logs and one exploratory borehole, crossed by the seismic line L507 (Fig. 4). We indirectly obtained seismostratigraphic information on Profile L103 from an interpreted seismic section that was recently published by ANP (Ferreira, 2013). In addition, information on the basement depth and rock types from six boreholes (Cordani et al., 1984) and new seismic sections published by Ferreira (2013) were taken into consideration to form an overall picture of the tectonic styles present in the basin. Based on these data, we recognized three different tectonic

patterns in the eastern, northern and western portions of the Parnaíba basin, which are described below.

We analyzed two well logs: the first (1FL) located to the west of the Transbrasiliano Lineament and the second (2BAC) located in the central portion of the Parnaíba basin (Fig. 4). Multi-parameter downhole logs included self-potential (SP), natural gamma ray (GR), electrical resistivity (IDL) and density (Rho). We were able to identify the major tectono-



sedimentary sequences in both wells (Fig. 7 and Table 1) primarily on the basis of the logging curve responses and their lithological associations, as described by Góes and Feijó (1994) and Vaz et al. (2007).

We correlated the well rock units with the seismic reflections using time-depth curves provided by ANP for this study (Fig. 8). The well-log correlation between well 1FL and line L509 and well 2BAC and line L305 (Fig. 4) allowed us to accurately identify the major sequence limits and volcanic sills. We performed the seismic interpretation using the seismic data and well ties. In addition, interval velocity was used in the time-to-depth conversions of the seismic sections.

### 3.5 Seismic and well-log correlation

We found no sedimentary strata of the Cambrian-Ordovician syn-rift unit (sequence I) in both 1FL and 2BAC wells. However, Petersohn (2010) described a 4,700-m-deep borehole (2PI in Fig. 4) that contained a stratigraphic interval at 3,000 m that they interpreted as the lowermost rift unit. In well 1FL, the low-grade metasedimentary rocks at 2250 m depth have low SP, low IDL and middle GR values (Fig. 7). Rb-Sr dating of these basement samples provided age of 670 Ma (Cordani et al., 1984). In contrast, the basement rocks have high values of SP, GR, IDL, and Rho in well 2BAC, revealing different geological units in central portion of the basin. The lower post-rift sequences II and III were deposited during the Silurian to Early carboniferous (Table 1). They are widely distributed across the entire basin (Fig. 3) and are almost 2,000 m thick in wells 1FL and 2BAC (Fig. 7). The SP values increase upwards, especially at the upper part of the sequence III. Shale layers have generally high GR value with bell-shaped curves, whereas box-shaped curves and low GR value and high IDL value match with volcanic rocks. The density log is available only in the last 1000 m of the well 2BAC (Fig. 7). The imaged siliciclastic layers have increased bulk density, whose Rho value ranges from 2470 to 2660 kg/m3, indicating intense compaction. Rho value higher than 2900 kg/m3 reveals at least two more than 50 m thick volcanic rocks in sequences II and III. The 110-m thickness of the shallow sequence IV occurs at the top of well 1FL. The thickness of this unit increases to 600 m to the west, toward the central part of the basin (well 2BAC in Fig. 7). The Sequence IV has upwards increasing SP value and middle to high GR value, whose both curves are finger-shaped or zigzag (well 2BAC in Fig. 7). Interdigital GR and SP curves with high value are related to intercalation of shales, siltstone, limestones, sandstones. The IDL curve is dominated by low resistivity lithology, but a few isolated IDL peaks could be associated with layers of mixed sands and anhydrite in a sabkha plain, as interpreted by Góes and Feijó (1994). The uppermost 530-m-thick Mesozoic sequence V is restricted to well 2BAC. Sandstones with fine intercalation of pelites and shales deposited in continental to shallow platform environments have high SP value, low GR value, and middle IDL. Unfortunately, IDL log shows no meaningful data in the first 250 m. The thicker sedimentary fill in well 2BAC indicates a considerably longer duration of subsidence, which lasted until the Cretaceous in the central part of the Parnaíba basin. In contrast, the deposition ceased in the Carboniferous at the eastern border of the basin (well 1FL in Fig. 7).

Interval velocities were calculated from time-depth curves (Fig. 8). The velocities rise downward from 2000 m/s to 4800 m/s due to the natural compaction of the sedimentary infill. Velocities up to 5100 m/s indicate the concentration of volcanic sills intercalated in the sequences II and III (Fig. 8). The correlation between the well lithologies and the seismic reflections



allowed us to characterize the four post-rift tectono-sedimentary sequences, as well the basement top and volcanic sills (Fig. 8). Although the syn-rift sequence I is not present in wells 1FL and 2BAC, its related seismic facies can be identified by a section of discontinuous, inclined, low amplitude reflections, enclosing two sets of high amplitude reflections related to volcanic rocks (Fig. 9a). The overlying sedimentary sequences II to V are characterized by parallel reflections with low to

5    middle amplitudes (Figs. 8 and 9). For this study, we used sixteen seismic lines, selected from available data (Fig. 4), to help characterize the tectonic styles in the Parnaíba basin (Figs. 9 and 10). The first style occurs in the eastern part of the basin, where seismic line L507 reveals a 120-km-wide and 4.5-km-deep rift zone. This rift zone is marked by a main graben and several secondary troughs (Fig. 8a). The eastern part of the graben represents the brittle reactivation of basement shear zones and coincides with the Transbrasiliano Lineament at the surface (Figs. 4 and 8a). The almost symmetric central graben is 25

10   km wide. Four major sedimentary sequences overlie the Precambrian basement (Table 1). A sedimentary sequence that is up to 2.5 km thick, which is correlated to the Cambrian-Ordovician Jaibaras Group (Oliveira and Mohriak 2003), fills the lower part of the rift zone (I in Fig. 8a). The upper sequences extend beyond the rift limits and were deposited between the Silurian and the Early Triassic (II, III, and IV in Fig. 8a). Large volcanic sills and dikes were emplaced in the basin fill. The entire volcanic and sedimentary package is deformed by normal and listric faults. Several faults, which reactivated and inverted

ancient features, were active until the Early Carboniferous (Morais Neto et al. 2013) and even formed Devonian-Carboniferous grabens at the eastern edge of the rift zone (Fig. 8a). In summary, the seismic lines indicate that the basement faults deform the entire sedimentary sequence and that the offsets related to these faults decrease from sequence I to sequence IV. These observations indicate that although tectonic activity slowed after the rifting period, faults continued to be reactivated during the post-rifting period.

The second tectonic style occurs in the central part of the Parnaíba basin, where a different tectonic scenario is observed in seismic line L304 (Fig. 8b). No graben-like structure occurs in this part of the basin, and the faults are less abundant, suggesting that deformation was less intense away from the Transbrasiliano Lineament. However, the compressional tectonic regime, which reactivated certain faults as reverse-related structures, is more evident in this region (e.g., the deep fault highlighted in the inset of Fig. 8b). These folds are consistent with the post-Carboniferous tectonic inversion that affected the

entire Parnaíba basin. From a stratigraphic perspective, the Paleozoic post-rift volcanic-sedimentary package is thicker in the central portion of the basin, where it is partially overlain by the upper Jurassic-Cretaceous sequence V.

        The third tectonic style occurs in the western portion of the Parnaíba basin and is evident in the seismic sections reported by Ferreira (2013). The imaged rift zone consists of two main asymmetric grabens separated by basement highs and listric faults (Fig. 9). These grabens are up to 100 km wide, are divided by smaller troughs, and trend NW-SE (Fig. 10), roughly

perpendicular to the NE-SW Transbrasiliano Lineament. As in the other basin regions (Fig. 8), large volcanic sills occur in different stratigraphic levels. However, the brittle deformation appears to be less intense in the western portion of the basin. The different internal basin architectures within the basin will be discussed in Section 5.



### 3.6 Simplified subsidence curves

The subsidence pattern of any sedimentary basin can be evaluated from plots of sediment age versus depth record in exploratory wells (Quinlan, 1987; Allen and Allen, 2005). Nevertheless, some corrections for sediment compaction, variations in paleobathymetry, and the isostatic amplification effects of the sedimentary load are necessary to obtain a reliable subsidence history (Steckler and Watts, 1978). Since providing a detailed modeling of basin subsidence is not the intent of the present paper, due to lack of sediment ages from the available wells, we constructed simplified subsidence curves of the Parnaíba basin based on three exploratory wells and the seismic line L103 (Fig. 11). These curves include two periods of accelerated subsidence rates, with amplitudes of 3500 m and 2000 m during the Cambrian-Ordovician and between the Silurian and Carboniferous, respectively. According to Quinlan (1987), the rapid syn-rift sediment accumulation certainly involves mechanical and thermal subsidence of the previously thermally uplifted lithosphere. During the rift phase (sedimentary sequence I), the subsidence rate was up to 47 m/Myr. After the rift became inactive, the Parnaíba basin experienced less rapid subsidence during the long period of subsequent cooling. The sedimentary sequences II and III (the Serra Grande and Canindé Groups) featured a sedimentation rate of approximately 22 m/Myr for 100 Myr (Fig. 10). From the Late Carboniferous onwards, the thermal subsidence slowed to a rate of 6.5 m/Myr, the cratonic basin stabilized, and the thin volcano-sedimentary sequences IV and V were deposited in the Jurassic and Cretaceous.

### 4 Magnetic and Gravity Joint Modeling

Based on a comparison of the potential field (Fig. 5) and seismic (Figs. 9 and 10) data, the pseudo-gravity and gravity lows clearly do not match the graben limits derived from seismic lines. This mismatch suggests that basement causative sources interfere with the anomaly patterns produced by the basin internal geometry. In this study, we performed a joint inversion of magnetic and gravity data along the seismic profiles to determine the contribution of each type of sources, i.e., the basement heterogeneities and the basin fill.

### 4.1 2D Joint modeling approach

For decades, many researchers have studied joint or cooperative inversion of two or more geophysical methods as a powerful tool to improve modeling (Santos et al., 2006; Zhou et al., 2015). Vozoff and Jupp (1975) were the first to invert jointly data sets derived from different geophysical methods in order to enhance inversion resolution of the models obtained separately from each method. A joint inversion of magnetic and gravity data, which are related to the same underlying geologic structures and thus may contain complementary information, can reduce the inherent nonuniqueness and the limitations of the inverse problem of individual geophysical methods (e.g., Lines et al., 1988; Haber and Oldenburg, 1997; Gallardo and Meju, 2003). Examples of joint inversion of magnetic and gravity data were presented and discussed by several studies, such as Fedi and Rapolla (1999), Gallardo et al. (2005), and De Castro (2011).





Using GM-SYS software, we performed the integrated approach on both residual RTP magnetic and gravity anomalies constrained by seismic data and well logs. The model geometry consists of the basement top, main sedimentary sequences and volcanic sills, all extracted from the seismostratigraphic interpretation tied with well logs. The densities of the basin fill were derived from density logs and density-velocity conversions using Gardner's formula, and magnetic susceptibilities

were initially obtained from average values reported in the literature (Telford et al., 1998). Table 2 presents the physical parameters of the modeled layers.

Before performing the joint modeling, we calculated the theoretical magnetic and gravitational effects of the seismic-derived models along lines L304 and L507 (Figs. 5 and 12). The flight elevation of 1100 m was properly incorporated into the anomaly calculation. Considering the magnetic susceptibilities and densities of the basin infill listed in Table 2 and

assuming that the basement possessed constant magnetic susceptibility of 0.041 SI units and density of 2750 kg/m3, the calculated anomalies related to the basin internal geometry (purple curves) do not match the residual reduced to pole magnetic and residual gravity anomalies (green curves). The difference between observed and calculated anomalies (red curves) represents the magnetic and gravity contribution of unknown basement rocks, which have no obvious expression in the seismic sections (Fig. 12). The basement causative sources yield high-amplitude anomalies, partially masking the

geophysical signatures of the basin internal architecture. Interestingly, the volcanic contribution to both potential field anomalies (blue curves) is almost negligible, which means that mapping of the magmatic events is difficult in the Parnaíba basin using airborne geophysical data collected at a flight height of 1100 m with a spacing of 6 km.

To create a joint gravity-magnetic model along the seismic profiles, we were required to introduce vertical prisms beneath the basin to correspond to the geotectonic units within the basement. In fact, a heterogeneous structural framework is

expected from the geological setting in the Jaibaras basin (Fig. 6), whose basement consists of several deformed Precambrian metasedimentary sequences and Neoproterozoic-Early Paleozoic syn-rift granite and volcanic bodies (Oliveira and Mohriak, 2003).

We applied a semi-automatic source detection method to guide the location of the vertical prism boundaries in the geophysical models. The chosen analytic signal technique computes discrete depth solutions at a moving spatial window

along magnetic and gravity profiles (Phillips, 1997). After fixing both the seismic basin internal architecture and prism geometry, a non-linear Marquardt inversion procedure, implemented in the GM-SYS inversion routine, iteratively obtained the densities and magnetic susceptibilities of the prisms (Table 2). The base of all prisms was fixed at a depth of 10 km. Figure 13 shows the final models along profiles L304, L507, L103, and L001, which were used to investigate different rift geometries and tectonic styles within the basin (Fig. 4). Profile L001 is the only profile not constrained by seismic data.

Nevertheless, to ensure that the modeling will provide a geologically reasonable result we used the final model of profile L507 as the initial model of the profile L001. In this way, based on the seismic data (Fig. 10a) we assumed the premise that the rifting style should be the same along the Transbrasiliano Lineament, changing only in lateral extent and depth. Thus, we performed the joint modeling of this profile by carefully modifying these variables until the calculated anomalies were adjusted to the observed data. We opted to not introduce volcanic rocks into this model due to the lack of seismic constraints.



## 4.2 Basin tectonic styles revealed by joint modeling

In the north-central part of the basin, where no rift-related structures were observed in the seismic lines, the gravity lows at both edges of profile L304 indicate the existence of less dense basement blocks rather than a steady westward increase in basin thickness (Fig. 13). The assigned magnetic susceptibilities (AMS) and densities vary between 0.00214 and 0.00385

(SI) and between 2594 and 2704 kg/m$^3$, respectively, within the basement. In the other tectonic domains of the basin, the rift zones are over 150 km wide and feature a main central symmetric graben and a set of secondary troughs (Figs. 10 and 13). Anchored by the Transbrasiliano Lineament at its eastern flank, the main graben crossed by lines L507 and L001 is up to 25 km wide and 4.5 km deep (Fig. 13). A low-density and low-AMS basement block either occurs to the west of or beneath the rift zone, as evidenced by profiles L507 and L001, respectively, and this block serves to conceal the gravity effect of the rift

geometry. The low-density crustal block (LDB) exhibits an average contrast of -375 kg/m$^3$, which may represent the low-grade metamorphic sequences and/or granite bodies of the Jaibaras basin structural framework (Fig. 6). No expression of the LDB is observed in the seismic profiles (Figs. 9a and 10a), making it difficult to identify its geological nature and internal geometry.

In profile L103, the main graben is characterized by a 60-km-wide, flattened and symmetric structure (Figs. 10 and 13).

The gravity low is shifted from the main graben to the SE due to a narrow low-density block with a density contrast of -91 kg/m$^3$. The WNW-ESE-trending master faults are oblique or orthogonal to those associated with the Transbrasiliano Lineament (Fig. 14). This peculiar rift architecture suggests an important change in the faulting mechanism within the basin. In this sense, the Transbrasiliano Lineament was reactivated in the Early Paleozoic by a brittle regime with a significant transtensional component, which controlled the NE-SW-elongated grabens. To the west, pure-shear extension appears to

have prevailed, forming grabens crosscutting the NE-SW-oriented basement fabric (metamorphic foliations and ductile shear zones).

## 5 Discussion

### 5.1. Mechanisms that generated accommodation space in the Parnaíba basin

The existence of Cambrian-Ordovician rifts in the Parnaíba basin poses several important questions. The first is whether

lithospheric extension or thermal subsidence was the driving mechanism for the growth of accommodation space during the initial phase in the formation of this large cratonic basin. This study is just beginning to shed light on these issues.

The Parnaíba basin exhibits three main structural styles (Figs. 9, 10 and 13). The first style is present in the eastern part of the basin and encompasses the NE-oriented elongated rift system formed by strike-slip brittle reactivation of the Brasiliano ductile shear zones. The second tectonic style is present in the central part of the basin, where the rift architecture

trends to WNW-ESE, perpendicular to the basement fabric (mainly shear zones and metamorphic foliations), as revealed by the magnetic and gravity lineaments (De Castro et al., 2014). Based on continental-scale geophysical and geotectonic maps



(e.g., Fairhead et al., 2003; Fuck et al., 2008; Brito Neves et al., 2014), no evident crustal or lithospheric discontinuities within the Parnaíba block contributed directly to the development of the rifting process. Finally, the third tectonic style is present in the northern part of the basin and is characterized by sparse or absent graben-like structures, which are not observed in the available seismic profiles (Figs. 9b and 10b). This region has been slightly affected by normal and reverse faulting (Fig. 9b), which was supposedly driven by the opening of the Equatorial Atlantic Ocean in the Early Cretaceous, according to Milani and Zalán (1998).

The geophysical data presented here indicate that during the Cambrian-Ordovician, crustal thinning became more focused within the Transbrasiliano shear zone (TB in Fig. 14) along en echelon pull-apart troughs by strike-slip faults and bounded to the west by the Parnaíba block (first tectonic style described above). In the vicinity of the TB, a 150-km-wide and 1200-km-long rift zone with a central symmetric graben formed and was anchored by this large shear zone along its eastern boundary (Figs. 9 and 10). This result clearly indicates that rift faults along the Transbrasiliano Lineament reactivated the mainly ductile shear zones, in agreement to what can been observed in the deep reflection profile (Daly et al. 2014). However, Daly et al. (2014) interpreted a lateral crustal discontinuity in the deep reflection profile as the expression of the Transbrasiliano Lineament at depth. This structure is located 100 km far away to the west from the Transbrasiliano Lineament originally mapped by previous studies (e.g., Cordani et al., 1984; Nunes, 1993; Fairhead et al., 2003; De Castro et al., 2014). Lineaments have been widely considered to be landscape features that maybe related to deep structures. The Transbrasiliano Lineament, defined approximately four decades ago, exhibits a clear surface expression. These features cross the Parnaíba basin and are associated with the graben system that we identified in the present study. The structures identified by Daly et al. (2014), which differ from the rift system in our study, may mark the boundary of crustal blocks, but these structures do not agree with the definition of the Transbrasiliano Lineament that is already established and recognized in the literature (Schobbenhaus et al., 1975). Therefore, we assume that the rift system we observed in the reflection seismic data is the deep expression of the Transbrasiliano Lineament.

To the west, the rift geometry changes and exhibits a wider and flattened central graben and a shorter rift zone (L103 in Fig. 9c). This modification of the tectonic style reveals the important role of the Transbrasiliano Lineament in controlling the development of the Cambrian-Ordovician rift systems (Figs. 14 and 15). Close to the lineament, oblique-slip crustal extension prevailed, generating an extensive rift system consisting of several NE-SW-oriented troughs. These graben-like features are located along the same structural trend and occur beyond the basin limits, as far as 960 km southwards (e.g., the Água Bonita and Piranhas basins, which are located ~1000 km to the south of the Parnaíba basin along the Transbrasiliano Lineament, Brito Neves et al., 1984). In contrast, more orthogonal stretching produced pure-shear extension, crosscutting the basement fabric (mainly shear zones and metamorphic foliations) in the central part of the Parnaíba basin (the second tectonic style defined above).



## 5.2 Post-rift tectonic activity

Subsequent tectonic episodes occurred during the post-rift phase, with brittle reactivation of Brasiliano shear zones. Reverse faults and folds imaged in the seismic sections deformed sedimentary sequences II and III, as well as the diabase sills (Fig. 9). Morais Neto et al. (2013) correlated this transpressional reactivation with the post-Devonian tectonic inversion reported by Destro et al. (1993), which gave rise to a km-scale gently plunging drag-shaped fold structure in the NE portion of the Jaibaras basin. Furthermore, later extensional reactivation of ancient shear zones especially deformed sedimentary sequence III, forming troughs close to the Transbrasiliano Lineament (Fig. 9a). The age relationships suggest that this reactivation postdates the Jurassic because the volcanic sills of the Mosquito formation (~200 Ma) are faulted and tilted. We hypothesize that the NW-SE-oriented extensional events are related to the Syn-rift II phase of the Cariri-Potiguar trend in the NE Brazil rift system, described by Matos (1992).

## 5.3 Subsidence evolution

In the Ordovician and Silurian, rifting ceased along the Transbrasiliano Lineament as a result of lithospheric thermal re-equilibration and contraction due to changes in the lithospheric stress field (Oliveira and Mohriak, 2003). The resulting negative buoyancy effect was likely the primary cause of the widespread and long-lived post-rift subsidence episodes that formed the broad saucer-shaped Parnaíba sag basin. For example, the simplified subsidence curves of the Parnaíba basin, extracted from three exploratory wells and a seismic section (Fig. 11), show two periods with accelerated subsidence rates, with amplitudes of 3500 m and 2000 m, during Cambrian-Ordovician times and Silurian to Carboniferous times, respectively. According to Quinlan (1987), the rapid syn-rift sediment accumulation involves subsidence of a previously thermally uplifted lithosphere. After the rift became inactive, the Parnaíba basin experienced slower subsidence during the long period of subsequent cooling. The sedimentary sequences II and III of the Serra Grande and Canindé Groups were deposited with a sedimentation rate of approximately 22 m/Myr for 100 Myr (Fig. 11). Analyzing cratonic basins in North America and West Siberia, Armitage and Allen (2010) noted that the slow extension of relatively thick continental lithosphere causes permanent, long-lived thermal subsidence in cratonic basins. In fact, a 40-km-thick crust, derived from receiver functions by Rosa et al. (2012) and 3D gravity inversion (De Castro et al., 2014), occurs in the western part of the Parnaíba basin, where the sag deposits are thicker and the thermal subsidence lasted for a longer time (2BAC curve in Fig. 11). This sedimentation rate is smaller than that during the rift phase, when mechanical subsidence was active in the Cambrian-Ordovician.

Following the Late Carboniferous, the thermal subsidence slowed to a rate of 6.5 m/Myr, the cratonic basin stabilized, and the thin volcano-sedimentary sequences IV and V were deposited in the Jurassic and Cretaceous. However, fault activity was not absent during this period of basin evolution. We have identified for the first time folds related to a post-rift inversion phase. This type of fold affected post-Permian to pre-Albian units and has already been identified in a deep seismic transect at the western margin of the basin (Daly et al., 2014).



## 5.4 Mapping pseudo-graben

The second question to be discussed involves the causes of the different trends in the pseudo-gravity and gravity lows, which do not spatially match the grabens identified in the seismic data (Fig. 9). The new seismic lines revealed not only the internal architecture of the basin but also a mismatch between the graben locations mapped from potential field and seismic data

(Fig. 14). In the Parnaíba basin, basement causative sources severely modify the magnetic and gravity patterns, concealing the anomalies directly related to the rift structures (Fig. 12). Analyzing the geological setting in the Jaibaras basin, we infer that the causative bodies responsible for the gravity lows beneath the Parnaíba basin could also be Neoproterozoic supracrustal sequences of low-grade metasedimentary rocks and anorogenic granites that surround the Cambrian-Ordovician Jaibaras rift (Fig. 6). Furthermore, the N-S-elongated gravity lows are parallel to the Araguaia Suture Zone along the western

edge of the basin (Fig. 14). In this region, low-density and low-grade metamorphosed successions, sparse ophiolite mafic and ultramafic rocks and granitic intrusions represent the Neoproterozoic Araguaia Belt (Moura et al., 2008). This belt formed as the result of the oblique collision of the Amazonian and São Francisco cratons, which also involved the Parnaíba block and the Tocantins and Borborema provinces (Alvarenga et al., 2000; De Castro et al., 2014) (black arrows in Fig. 14). The metasedimentary sequences were thrusted over the eastern edge of the Amazonian Craton, forming thick packages of

low-density rocks in the upper crust.

Based on the N-S-oriented gravity lows, we hypothesize that these supracrustal sequences were also deposited to the east of the Araguaia Suture Zone within the Parnaíba block. These sequences in places assume a NW-SE orientation and are limited by the Transbrasiliano Lineament to the south and east (Fig. 14). In fact, comparing basement rocks drilled in exploratory wells in the Parnaíba basin (available in Cordani et al., 1984) with the interpreted residual gravity map (Fig. 14),

we observe a close correlation between the negative anomalies and the low-grade metamorphic rocks and felsic intrusive rocks. Cordani et al., (1984) described these rocks as phyllites, quartzites and syenites. Rb-Sr dating has provided ages that range from 670 to 504 Ma. At that time, the Brasiliano-Pan African orogeny was active, leading to the overall amalgamation of West Gondwana and possibly causing an elongated supracrustal strip to be thrusted over the Parnaíba block, similar to the Araguaia Belt, which was emplaced onto the Amazonian Craton to the west (Fig. 15). We interpret the low-grade

metamorphic rocks and felsic intrusive rocks to be the most likely candidates for the low-density bodies (LDB) that have been modeled within the basement (Fig. 13).

Between the Ediacaran and the Ordovician, the widespread post-orogenic extensional tectonic regime (blue arrows in Figs. 14 and 15) led to the onset of post-orogenic magmatism accompanied by the intrusion of granite bodies and continental rifting along the Brasiliano shear zones, particularly along the Transbrasiliano Lineament (TB in Figs. 14 and 15) (Brito

Neves et al., 1984; Oliveira and Mohriak, 2003; Cordani et al., 2013). Examples of post-orogenic granite intrusions occur along both the NE and SW edges of the Parnaíba basin in the vicinity of the TB. Additional granite bodies, emplaced along the TB, are likely concealed beneath the basin, contributing locally to pseudo-gravity and gravity negative anomalies, in addition to the supracrustal sequences (Fig. 15).



## 5.5 Broad implications for other basins in West Gondwana and elsewhere

The evolution of the Parnaíba basin has implications for other basins in both South America and Africa. The questions associated with the development of these basins appear to be related, and their answers have important implications for the origin and development of the long-lived and extensive Parnaíba cratonic basin.

The rifting process related to the breakup of Pangea in the Jurassic and Cretaceous does not appear to have played a major role in the generation of accommodation space in the Parnaíba basin. This process, however, has been described as the primary tectonic event responsible for several sedimentary basins along the eastern margin of South America (De Castro et al., 2007; De Castro and Bezerra, 2015). Therefore, several basins in northeastern Brazil exhibit both Paleozoic and Jurassic-Cretaceous sedimentary units (e.g., Araripe and Tucano-Jatobá basins, Fig. 2). In these cases, the Paleozoic units have been

described as undeformed pre-rift units generated by a sag deposition (e.g., Matos, 1992; Assine, 2007). This type of hypothesis is based on the assumption that the deposits in the Parnaíba basin, which encompasses the major Paleozoic sequence in the region, were formed through sag-related deposition. This study indicates that at least two rift phases preceded the sag stage. In addition, the geophysical data from our study shows that the sag-related deposition was also controlled by the reactivation of rift faults (Fig. 9). In a few basins, such as the Araripe and Tucano-Jatobá basins, Paleozoic

sedimentary units are intensively deformed, and this deformation has been attributed to Mesozoic rifting (Magnavita et al., 1994; Assine, 2007; Kuchle et al., 2011). However, Early Paleozoic rifting episodes may also have affected these basins, which have been subjected to rifting in the Early Paleozoic and in the Jurassic-Cretaceous. This hypothesis should not be ruled out and should be investigated in further studies.

## 6 Conclusions

The Parnaíba basin is a large, sag-type cratonic basin with a saucer shape and roughly circular shape. Its history of long-term accumulation of terrestrial and shallow-water marine sediments started following the final assembly of the Amazon–West African block in an overall collisional scenario involving the Araguaia and Transbrasiliano megashear zones. Post-orogenic tectonic inversion occurred during the Late Neoproterozoic and Early Paleozoic, forming a set of troughs. The sag-related deposition in the failed rift system was similar to that of many coeval cratonic basins scattered throughout North America,

Africa and South America.

A large geophysical dataset, involving airborne potential field, seismic and well log data, was used to map the buried graben-like structures to shed light on the driving mechanisms of the prolonged subsidence of the Parnaíba cratonic basin. The combined analysis and joint modeling of the geophysical data revealed a complex basement framework that includes elongated Neoproterozoic intraplate aulacogenic-type basins. These approximately N-S-oriented low-grade metasedimentary

strips were formed in the ancient Parnaíba block during the oblique continental collision between the Amazonian and São Francisco cratons. The basement heterogeneities strongly interfere in the magnetic and gravity anomaly patterns; thus, the basin internal geometry can only be correctly mapped when the potential field data are constrained by seismic profiles and



well logs. Additionally, anorogenic granites also contribute to masking the gravity effects of the basin in the vicinity of the rift zone. Similar granitic intrusions were emplaced in the Jaibaras rift to the NE of the Parnaíba basin.

The basin framework can be divided into three main tectonic styles based on the distribution of the graben-like features revealed by seismic data. In the easternmost tectonic domain, the NE-SW-trending Brasiliano shear zones controlled the

rifting process, primarily along the crustal boundary between the Parnaíba block and the Neoproterozoic Borborema Province. An elongated rift zone up to 150-km-wide was formed by brittle reactivation of old shear zones in an oblique-slip crustal extension setting. The 30-km-wide symmetric central graben is anchored by the Transbrasiliano shear zone. To the west, in the south-central tectonic domain, the main axis of the rift zone trends NW-SE, orthogonal to regional metamorphic foliations and ductile shear zones in the Parnaíba block. The central graben is larger and deeper and exhibits a flat bottom.

Finally, the northern tectonic domain exhibits no seismic evidence of Cambrian-Ordovician rifting, and the post-rift deformation within the basin is restricted to rare subtle normal and reverse faults.

The simplified subsidence curves from boreholes and seismic lines reveal periods of accelerated subsidence in both syn- and post-rift phases. During the rifting phase, the combined mechanical and thermal subsidence rapidly generated accommodation space, resulting in a sedimentation rate of 35 m/Myr. After the rifting became inactive in the Late

Ordovician, the subsidence rate decreased, although it remained relatively high during the Silurian and Carboniferous. Mantle flow regime may provide a plausible process for cratonic subsidence, but geological and geophysical evidence precludes this type of mechanism in the Parnaíba basin. Additionally, secondary mechanisms affected the prolonged thermal subsidence established by changing tectonic stresses associated with plate motion and mantle dynamics. Following the Late Carboniferous, the thermal subsidence slowed, the cratonic basin stabilized, and thin volcanic-sedimentary sequences were

deposited during the Jurassic and Cretaceous. The deposition of these units is likely related to lithospheric deformation during the opening of the Equatorial Atlantic between the latest Jurassic and the Early Cretaceous.

*Acknowledgements*. We acknowledge the Brazilian Agency of Petroleum and Gas (Agência Nacional de Petróleo, Gás Natural e Biocombustíveis - ANP), which provided the seismic, borehole, gravity, and magnetic data used in our study.

Brazilian oil company Petrobras, as part of the Transbrasiliano Project, and INCT- Estudos Tectônicos funded this work. The authors are grateful to the Conselho Nacional de Desenvolvimento Científico e Tecnológico (CNPq) for their research grants.

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





**Tables**

Table 1. Chronostratigraphic chart illustrating various divisions of the Phanerozoic stratigraphy of the Parnaíba basin.

| Chronostratigraphic scale | | Lithostratigraphy | | | | Lithologic column |
|---|---|---|---|---|---|---|
| Age (Ma) | Period | Phase | Sequence | Group | Formation | |
| 94 - 200 | Jurassic - Cretaceous | Sag / South Atlantic rift | V | - | Mosquito, Pastos Bons, Sardinha, Corda, Itapecuru | sandstones, pelites and shales. basalt |
| 223 - 310 | L. Carboniferous – E. Triassic | Sag | IV | Balsas | Piauí, Pedra do Fogo, Motuca, Sambaíba | shales, siltstone, limestones, sandstones |
| 334 - 400 | M. Devonian – E. Carboniferous | Sag | III | Canindé | Itaim, Pimenteiras, Cabeças, Longá, Poti | siltstones, shales and sandstones |
| 400 - 443 | Silurian – E. Devonian | Sag | II | Serra Grande | Jaicóis, Tianguá, Ipu | fluvial and deltaic sandstones, pelites |
| 500 - 527 | Cambrian | Rift | I | - | Jaibaras | conglomerates, sandstones, phyllites and shales |
| > 540 | Precambrian Basement | | | | | |

5 Table 2. Bodies used in 2D geophysical profile models. AMS: assigned magnetic susceptibility (mathematical representation that incorporates both magnetic susceptibility and a remnant component, if applicable).

| Stratigraphic Sequence | Geological unit | Lithology | Density (kg/m$^3$) | AMS (SI) |
|---|---|---|---|---|
| V | Mearim | sandstones, pelites and shales | 2250 | 0.0001 |
| | Mosquito | basalt | 2690 | -0.006 – 0.055 |
| IV | Balsas | shales, siltstone, limestones, sandstones | 2400 | 0.0001 |
| III | Canindé | siltstones, shales and sandstones | 2450 | 0.0001 |
| II | Serra Grande | fluvial and deltaic sandstones, pelites | 2530 | 0.0001 |
| I | Jaibaras | conglomerates, sandstones, phyllites and shales | 2570 | 0.0001 |
| Precambrian Basement | - | Unknown basement rocks | 2569 – 2792 | -0.010 – 0.041 |




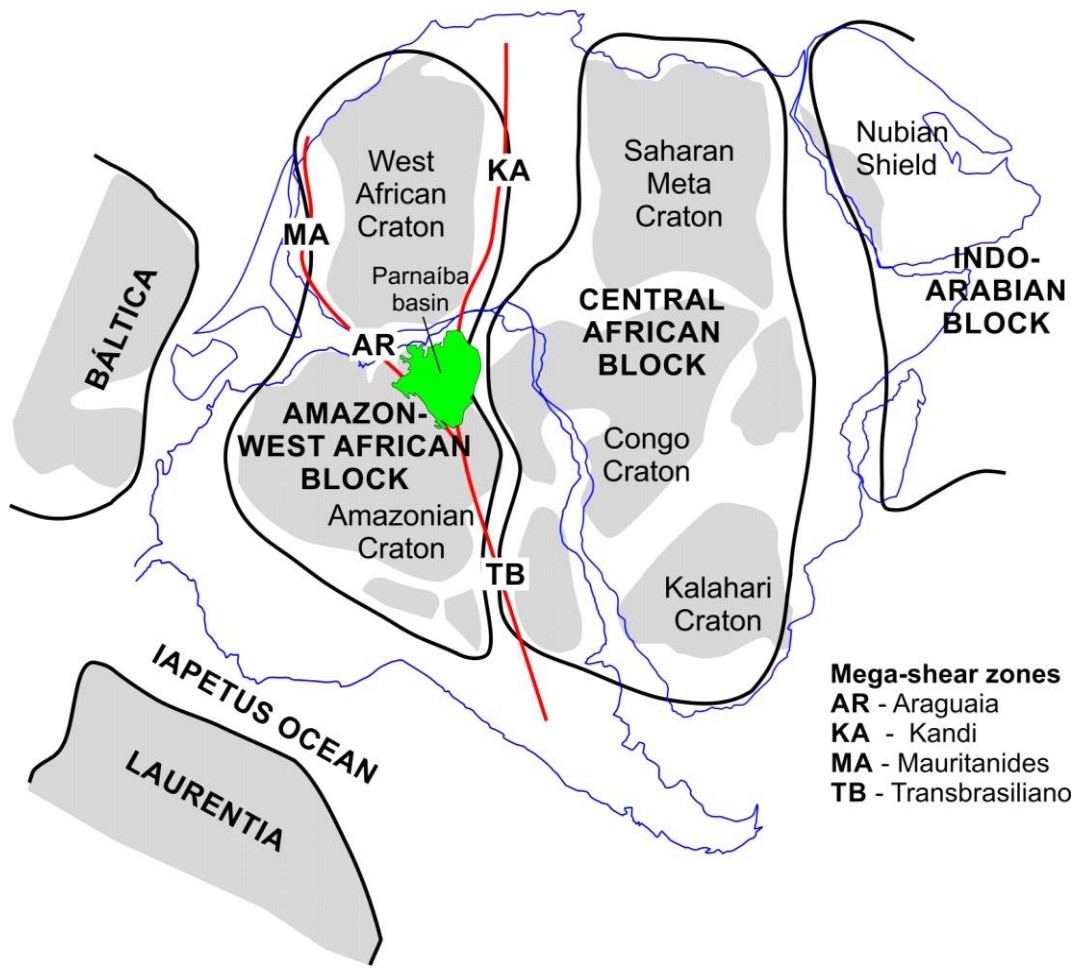

Figure 1. Reconstruction of West Gondwana at ca. 540 Ma, showing major crustal domains and continental shear zones discussed in the text. Key: red lines represent shear zones; blue lines represent continents; thick black lines represent West Gondwana continental blocks. Adapted from Cordani et al. (2013) and Stampfli et al. (2013).





Figure 2. Schematic geological map of NE Brazil showing the Parnaíba basin (blue line) and its Precambrian basement structures (red lines). Gray areas correspond to Mesozoic to recent sedimentary covers. Labeled dark gray areas correspond to exposures of Cambrian-Ordovician troughs and their supposed prolongation beneath the sag basin: 1 – Jaguarapi; 2 – Jaibaras; 3 – Cococci; 4 – São Julião; 5 – São Raimundo Nonato; 6 – Correntes; 7 – Monte do Carmo. Brasiliano shear zones: AR – Araguaia; PA – Patos; PE – Pernambuco; SP – Senador Pompeu; TB – Transbrasiliano; TG – Tentugal. Basins cited in text: Araripe (Ar) and Tucano-Jatobá (Tu-Ja).



Figure 3. Simplified geological map of the Parnaíba basin. Stratigraphic units: 1 – Silurian; 2 – Devonian-Carboniferous; 3 – Permian-Triassic; 4 – Jurassic; 5 – Cretaceous; 6 – volcanic rocks; 7 – Early Cretaceous São Franciscana basin; 8 – Cenozoic to recent sedimentary cover. Brasiliano shear zones: AR – Araguaia; PA – Patos; PE – Pernambuco; SP – Senador Pompeu; TB – Transbrasiliano; TG – Tentugal.





Figure 4. Location of the airborne gravity and magnetic survey (gray area), seismic profiles (blue and black), and drilling wells (green). The seismic lines in black were extracted from Ferreira (2013).




Figure 5. Residual reduced-to-pole magnetic (a), gravity (b), and pseudo-gravity (c) anomaly maps of the Parnaíba basin. Main pseudo-gravity and gravity lows are highlighted in (d). Brasiliano shear zones: AR – Araguaia; PA – Patos; PE – Pernambuco; SP – Senador Pompeu; TB – Transbrasiliano; TG – Tentugal. Labeled anomalies are discussed in the text. Magnetic and gravity anomalies along seismic lines L304 and L507 are shown in Figure 12.




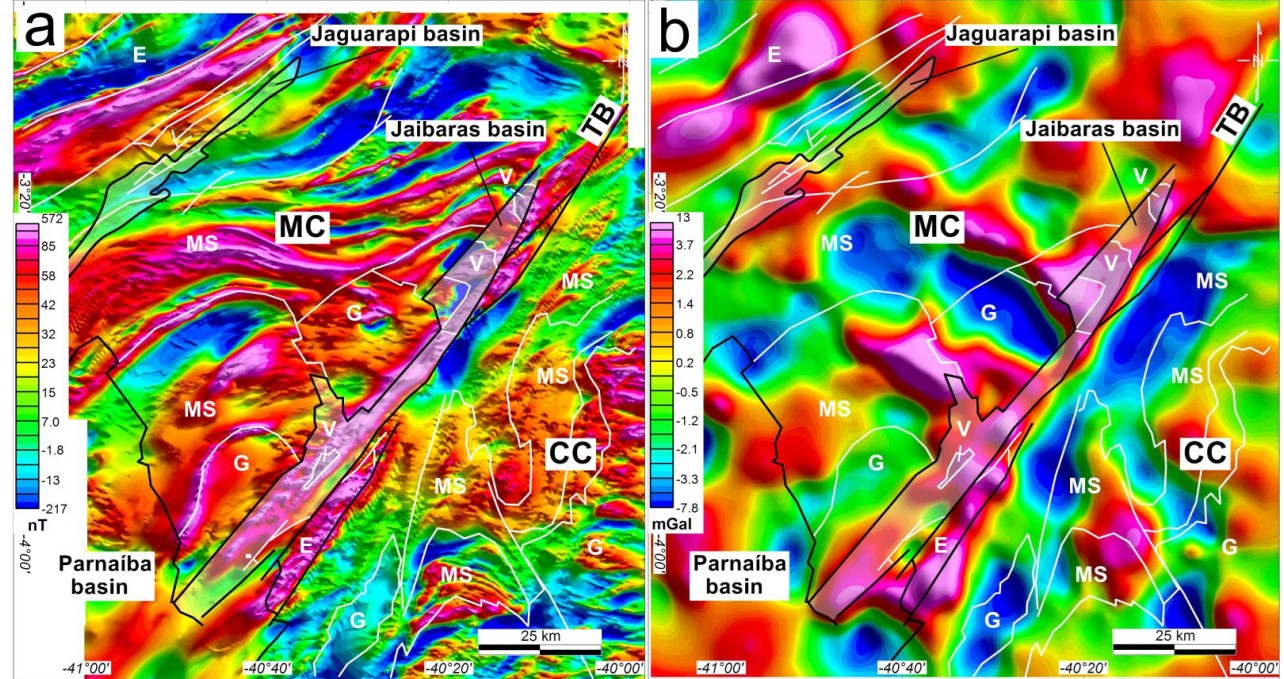

Figure 6. Residual reduced-to-pole magnetic (a), and gravity (b) anomaly maps of the Jaibaras basin. Geological contacts were extracted from Cavalcante *et al.* (2003). Tectonic domains: MC – Médio Coreaú; CC – Ceará Central. Geological units: E – granulites and eclogites; G – granites; MS – low-grade metasedimentary units; V – volcanic rocks. TB – Transbrasiliano Lineament.



Figure 7. Detailed well logging curves of exploratory wells 1FL (left) and 2BAC (right). The well-tops of the main tectono-sedimentary sequences were defined based on the reference lithostratigraphic chart described by Góes and Feijó (1994).





**Tectono-sedimentary sequences**
II - Silurian - E. Devonian
III - M. Devonian - E. Carboniferous
IV - L. Carboniferous - E. Triassic
V - Jurassic - Cretaceous

Figure 8. Time-depth conversion and well-seismic tie of wells 1FL and 2BAC and seismic lines L509 and L305. Data locations in Figure 4.





Figure 9. Seismic sections of lines L507 (a), L304 (b) and L103 (extracted from Ferreira (2013) - c), showing different structural settings within the Parnaíba basin. Inset in Line L304 shows reverse-related structures affecting Late Carboniferous to Early Triassic sequences.





Figure 10. Interpreted seismic sections showing different tectonic styles in the eastern (a), northern (b) and central (c) regions of the Parnaíba basin. The seismic lines in (c) were extracted from Ferreira (2013).





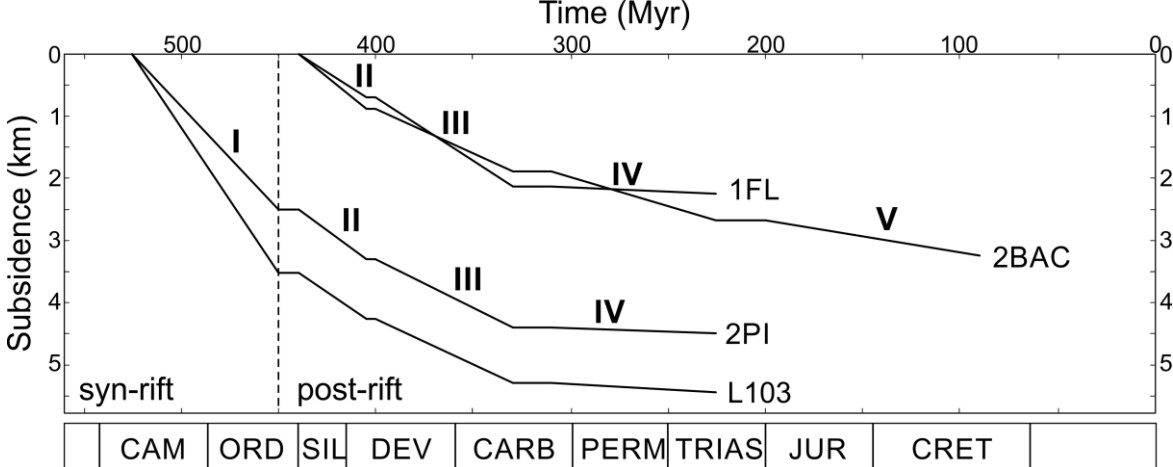

Figure 11. Sediment thickness as a function of sediment age for wells 1FL, 2BAC, and 2PI and seismic line L103 (at the distance of 40 km – Fig. 9). Labels I to V indicate stratigraphic sequences described in the text.





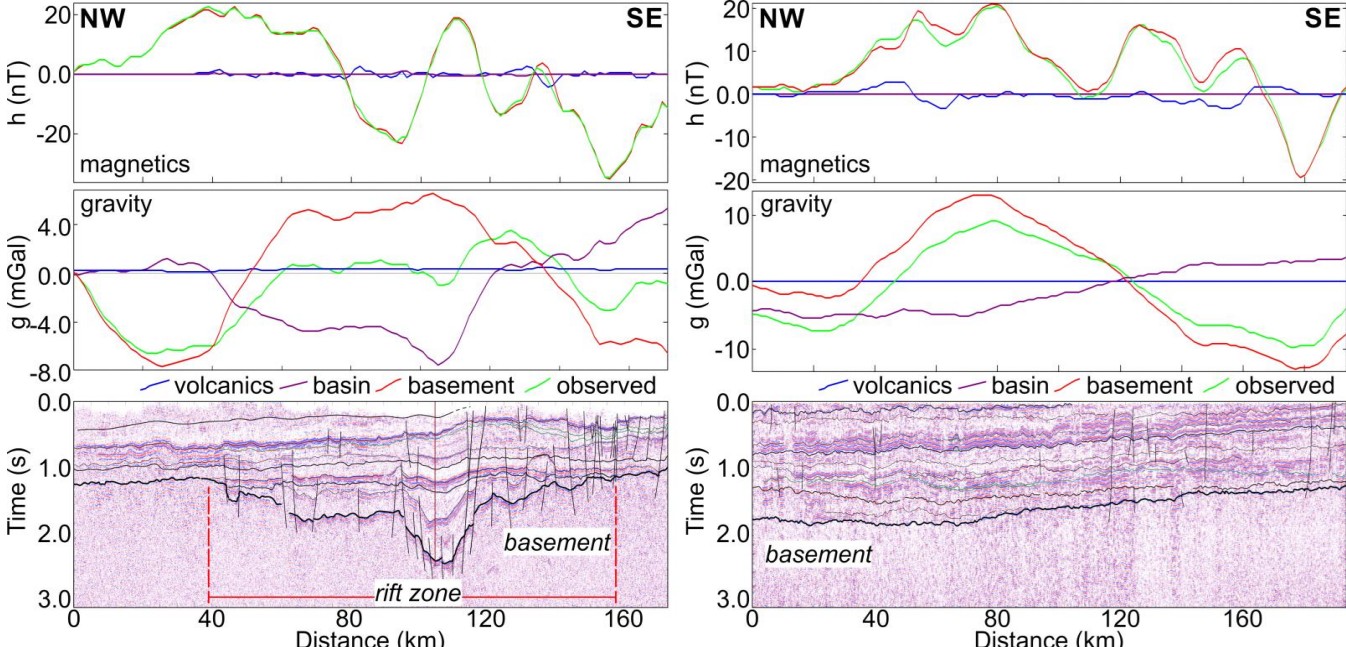

Figure 12. Observed and calculated residual reduced-to-pole magnetic and gravity anomalies along seismic lines L507 (left) and L304 (right) (location in Figure 6). The basin anomalies are calculated from the seismic basin geometry. The theoretical basement curves were obtained by subtracting the basin anomalies from the observed data.

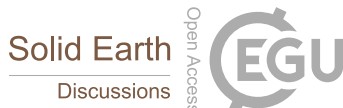



Figure 13. Joint modeling of residual reduced-to-pole magnetic (a), and gravity (b) anomalies along seismic profiles. Final adjusted magnetic (c), and gravity (d) models. HDB: high-density crustal block; LDB: low-density crustal block.





Figure 14. Simplified tectonic domains of the Parnaíba basin, showing outcropping marginal troughs (dark grey areas), seismic derived Cambrian-Ordovician rift zones (green areas) and gravity lows (light grey areas). Arrows indicate Brasiliano collisional (black) and post-orogenic extensional (blue) stresses. Red triangles indicate low-grade metasedimentary rocks drilled in exploratory wells. SFB: São Franciscana basin. Brasiliano shear zones: AR – Araguaia; PA – Patos; PE – Pernambuco; SP – Senador Pompeu; TB – Transbrasiliano; TG – Tentugal.





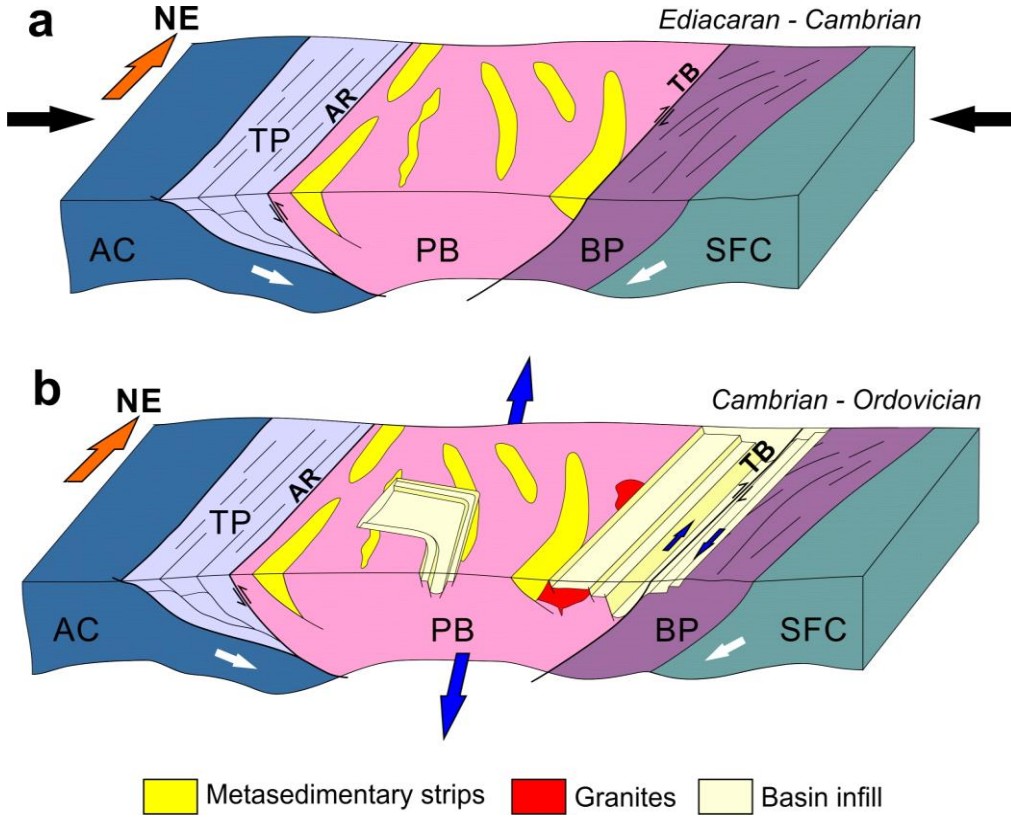

Figure 15. Schematic reconstruction for the Neoproterozoic-Early Paleozoic, showing the collision of the Amazon-West African block between the Araguaia and Transbrasiliano shear zones (a) and subsequent post-orogenic rifting (b). Arrows indicate Brasiliano collisional (black) and post-orogenic extensional (blue) stresses. AC - Amazonian craton; SFC - São

5 Francisco craton; PB - Parnaíba block; TP - Tocantis province; BP - Borborema province; AR - Araguaia suture zone; TB - Transbrasiliano shear zone.