# Peer review of "Geophysical evidence of pre-sag rifting and post-rifting fault reactivation in the Parnaíba basin, Brazil"

_Solid Earth, 2016_

## Referee Comment (RC1) · Anonymous Referee #1 · 11 Feb 2016

De Castro et al. interpret different geophysical observations, i.e. gravity, magnetic, seismic and borehole data, to study the tectonic mechanisms connected to the Parnaiba basin, a large intracratonic basin in the South American platform.

Although outside my main field of expertise, I found the paper well written and clearly exposing the geodynamic context and open questions about the formation and evolution of this, as well as of others intracratonic basins. In any case, I will mostly comment on the methodological aspects of data processing and interpretation. All the methodologies used are appropriate and their integrated geological interpretation is also done with critical reasoning. There are, however, few points where I think the authors should pay more attention (see detailed comments below).

Detailed review
[Figure]

Previous studies based on magnetic and gravity interpretation by same authors (De Castro et al. 2014) have been carried out. The outcome of that study was the inference of concealed grabens. In view of the limited resolving power of potential field data, here the authors add new geophysical observations to better constrain their interpretation. Specifically, seismic and well data have been used.

The pre-processing of geophysical data, partly done by the Brasilian Petroleum Agency (ANP), is appropriate and well explained.

Regional anomalies have been filtered out and pseudogravity conversion of magnetic map has been obtained. The authors mention the difference between the pseudogravity map obtained in this study and the one by their previous work (De Castro et al. 2014) and discuss possible reasons for it.

The comparison of gravity and pseudogravity maps in Fig. 5 is, in my opinion, a bit confusing. The two maps reflect, of course, two different physical properties and have different wavelength content. More comment on the possibility of long-wavelength artifacts in the pseudo-gravity map should be added. What kind of assumption has been made outside the survey area?

About the seismic lines used, I have a bit of perplexity on the identification of three different tectonic regimes based solely on the sections here presented. In particular, it is hard to compare the L507 and L304 seismic lines with the Ferreira's one (Line 303, bottom panel in Fig. 9 and Fig. 10c) (by the way link to Ferreira's study is missing). In particular, the quality of the interpreted seismic sections in Fig. 10c and 9 is low and I feel that reliability of the interpretation rely more on trust than in a real assessment of data. Some words of caution and more critical assessment of seismic sections, corroborated by better illustrations and proper references, should help the readers to build their own opinion.

My main concern regards, however, the joint gravity and magnetic inversion along the seismic sections. I am not sure about the added value of this analysis. The authors

should explain if the performed inversion is 2-D or 3-D, describe what kind of relationship between physical properties and what ranges of parameters (beside the basement) have ben tested, and test (and discuss) trade-off with uncertainties of seismic horizons. This way, the authors should have a better assessment of the outcome and limitations of the inversion.

Minors

Page 10, row 18-19. Based only on interpretation of seismic sections, this sentence sounds a bit too strong. I advise to modify it.

Page 10, row 21 and row 24: It is Fig. 9b and not 8b.

References: Ferreira 2013 is missing.

---

## Referee Comment (RC2) · Anonymous Referee #2 · 1 Mar 2016

Dear Editor of Solid Earth, please find below my review of the paper Geophysical evidence of pre-sag rifting and post-rifting fault reactivation in the Parnaíba basin, Brazil by De Castro et al., submitted to Solid Earth.

De Castro and co-workers analysed a multidisciplinary dataset (air-magnetic and gravity data, seismic line data, well log data) to reveal the tectono-sedimentary evolution of the intracratonic Parnaiba basin, one of the largest Palaeozoic basin in NE Brazil and south America. The subject of intraplate deformation, basin formation and multiple reactivation of inherited structures in cratonic areas is certainly worth to document, with important implications for both academic and industrial purposes. The value of the present paper is that different geophysical data are integrated and

discussed in the framework of the overall tectonic evolution of NE Brazil. Furthermore, the Parnaiba basin is a very large basin which lies in a remote area where, evidently, the systematic acquisition of large filed-based datasets is still in progress. In this view, the present paper provides further insights to understand the overall long-lasting tectono-sedimentary evolution of NE Brazil. Despite geophysical data analyses and modelling presented in this paper is not my primary expertise, I found the figures clear and explicative. The authors propose a multistage model in which sag basin sedimentation was preceded by an Early Palaeozoic rift stage, possibly linked with brittle reactivation of previously formed crustal-scale lineaments. The proposed model is sound and supported by data presented. For these reasons, I recommend the paper for publication in SE. I have found, however, some typos throughout the manuscript, which are highlighted in the annotated PDF copy of the manuscript together with some comments and suggestions (mainly on terminology used). Kind regards.

Please also note the supplement to this comment:
http://www.solid-earth-discuss.net/se-2016-21/se-2016-21-RC2-supplement.pdf

[Figure]

**Supplement:**

[revised manuscript text omitted]

---

## Author Comment (AC1) · 8 Mar 2016

1. The comparison of gravity and pseudogravity maps in Fig. 5 is, in my opinion, a bit confusing. The two maps reflect, of course, two different physical properties and have different wavelength content. More comment on the possibility of long-wavelength artifacts in the pseudo-gravity map should be added. What kind of assumption has been made outside the survey area? Reply: We clarified the meaning of the pseudo-gravity anomalies, the differences between them, the additional comments on the gravity anomalies and the assumptions we made outside the survey area in Page 7, Lines 23-32.

2. About the seismic lines used, I have a bit of perplexity on the identification of three

different tectonic regimes based solely on the sections here presented. Reply: We now make it clear that our purpose was not describe in detail tectonic evolution of the Parnaíba Basin, but to point out that there are significant variations in the structural style, reactivation patterns and trends of main structures based on the existing geophysical data (Page 11, Lines 11-15). These differences indicate that the tectonic-sedimentary evolution of the basin should be described as diachronic in space and time.

3. In particular, it is hard to compare the L507 and L304 seismic lines with the Ferreira's one (Line 303, bottom panel in Fig. 9 and Fig. 10c) (by the way link to Ferreira's study is missing). In particular, the quality of the interpreted seismic sections in Fig. 10c and 9 is low and I feel that reliability of the interpretation rely more on trust than in a real assessment of data. Some words of caution and more critical assessment of seismic sections, corroborated by better illustrations and proper references, should help the readers to build their own opinion. Reply: We received the seismic data of Profile 103 and the parallel seismic lines. We interpreted and included them in Figures 9 and 10. The illustrations of Figures 9c and 10c are now improved. The missing reference has also been included.

4. My main concern regards, however, the joint gravity and magnetic inversion along the seismic sections. I am not sure about the added value of this analysis. The authors should explain if the performed inversion is 2-D or 3-D, describe what kind of relationship between physical properties and what ranges of parameters (beside the basement) have been tested, and test (and discuss) trade-off with uncertainties of seismic horizons. This way, the authors should have a better assessment of the outcome and limitations of the inversion. Reply: We clarified the 2D joint modeling approach, describing its contribution to mapping the buried rifts isolated by intra-basement causative sources (Pages 12, Lines 15-20). We also mentioned which parameters we investigated and the uncertainties of seismic horizons (Page 13, Lines 6-8).

5. Page 10, row 18-19. Based only on interpretation of seismic sections, this sentence sounds a bit too strong. I advise to modify it. Reply: We rewrote the sentence to

suggest that fault reactivation occurred during the post-rifting period (Page 10, Lines 29-31).

6. Page 10, row 21 and row 24: It is Fig. 9b and not 8b. Reply: We corrected the text.

7. References: Ferreira 2013 is missing. Reply: We included this missing reference.

Anonymous Referee # 2 I have found, however, some typos throughout the manuscript, which are highlighted in the annotated PDF copy of the manuscript together with some comments and suggestions (mainly on terminology used). Reply: All annotated revision were corrected in the text.

Please also note the supplement to this comment:
http://www.solid-earth-discuss.net/se-2016-21/se-2016-21-AC1-supplement.pdf

TB: Transbrasiliano
     Lineament
**V**: volcanic sills

Basement

*Main graben*

*Rift zone*

a

Tectono-sedimentary sequences
I - Cambrian (rift phase)

[revised manuscript text omitted]

The pseudo-gravity transformation attenuated the high-frequency content of the magnetic anomalies (Figs. 5a and c). It is noteworthy that pseudo-gravity lows coincide with prolongations of the Cambrian-Ordovician troughs, which are partially

25   exposed at the eastern and southeastern edges of the Parnaíba basin (Fig. 2). Based on this pseudo-gravity anomaly pattern, we interpret the negative anomalies as main candidates for the magnetic response of the concealed rifts (Fig. 5d). The different gravity and pseudo-gravity signatures indicate that the assumption of same source causing gravity and magnetic anomalies is violated in the survey area. A possible explanation is the difference between the contrasts of the physical properties combined with decay rates of the gravitational and magnetic fields. Apparently, the pseudo-gravity anomalies

30   reflect the contrast of the magnetic susceptibilities between sedimentary and crystalline rocks, whilst the gravity anomalies are strongly influenced by deeper sources within the basement. This hypothesis is tested by the 2D magnetic-gravity joint modeling in section 4.

**3.3 Magnetic and gravity anomaly patterns in the Jaibaras basin**

[revised manuscript text omitted]

The three tectonic styles described above based on the existing geophysical data indicate that the tectonic-sedimentary evolution of the basin is diachronic in space and time. It follows that pre-sag rifting and post-rifting deformations are not homogenous across the basin. For example, preexisting ductile fabric controls fault reactivation along the Transbrasiliano Lineament, but fault reactivation is less evident away from the lineament. In addition, reverse faulting, not commonly

15   observed in the Transbrasiliano Lineament area, clearly occurs in the central part of the basin.

**3.6 Simplified subsidence curves**

[revised manuscript text omitted]
 thanks two anonymous reviewers and Solid Earth Special Editor Frederico Rossetti for their comments and suggestions, which greatly improved our work. 
[revised manuscript text omitted]